# Direct probing of single-molecule chemiluminescent reaction dynamics under catalytic conditions in solution

Ziqing Zhang[1], Jinrun Dong [1], Yibo Yang[1], Yuan Zhou[1], Yuang Chen[1], Yang Xu[1] & Jiandong Feng [1,2] ✉

Chemical reaction kinetics can be evaluated by probing dynamic changes of chemical substrates or physical phenomena accompanied during the reaction process. Chemiluminescence, a light emitting exoenergetic process, involves random reaction positions and kinetics in solution that are typically characterized by ensemble measurements with nonnegligible average effects. Chemiluminescent reaction dynamics at the single-molecule level remains elusive. Here we report direct imaging of single-molecule chemiluminescent reactions in solution and probing of their reaction dynamics under catalytic conditions. Double-substrate Michaelis–Menten type of catalytic kinetics is found to govern the single-molecule reaction dynamics in solution, and a heterogeneity is found among different catalyst particles and different catalytic sites on a single particle. We further show that single-molecule chemiluminescence imaging can be used to evaluate the thermodynamics of the catalytic system, resolving activation energy at the single-particle level. Our work provides fundamental insights into chemiluminescent reactions and offers an efficient approach for evaluating catalysts.

Chemical reactions generate products as a result of electron transfer during the forming and breaking of chemical bonds between atoms, which is accompanied by physical phenomena, such as heat conversion and light generation. Observation of chemical reactions can be achieved by monitoring the changes of reactant, the formation of products or probing the accompanied physical effects, which are usually viewed at the ensemble level[1]. Although some studies observing individual chemical reactions have been reported[2,3], direct probing of chemical reaction positions and their dynamics at the single-molecule level in solution has been a long-standing challenge in chemistry. Due to the faintest physical signals from individual reactions, most of the available single-molecule chemistry approaches measure the property of chemical substance from a reaction output, such as the scanning probe-based approach measuring the conductivity of the substance[4] and single-molecule fluorescence microscopy monitoring the fluorescence of

the transform[5–10]. We recently achieved the direct observation of single-molecule reactions of a specific chemiluminescence phenomenon (electrochemiluminescence) near the electrode surface[11,12]. However, general single-molecule chemiluminescent reaction events initialized by the free diffusion and the stochastic collision of molecules deep in solution can result in large uncertainties of the reaction location and the reaction time. Revealing such chemiluminescent reaction dynamics at the single-molecule level remains challenging.

In this study, we report the direct imaging of single-molecule chemiluminescent reactions in solution and probing of their reaction dynamics under catalytic conditions. Our experiment directly addresses the reaction itself by monitoring the chemiluminescence from the reaction without any background caused by laser and autofluorescence, which is beneficial for probing catalysis.

[1]Laboratory of Experimental Physical Biology, Department of Chemistry, Zhejiang University, 310058 Hangzhou, China. [2]Research Center for Quantum Sensing, Research Institute of Intelligent Sensing, Zhejiang Lab, 311121 Hangzhou, China. ✉e-mail: jiandong.feng@zju.edu.cn

## Results and discussion

Chemiluminescence is a light emission process, in which a single photon is emitted from a single-molecule chemical reaction, featuring advantages of ultralow background and high sensitivity in imaging[13,14] and immunoassay[15,16]. A classical chemiluminescence system is used in our experiments (Fig. 1a, b): luminol (5-amino-2,3-dihydro-1,4-phthalazinedione, $LH^-$) reacts with hydrogen peroxide ($H_2O_2$) under the catalysis of horseradish peroxidase (HRP).

To understand the enzymatic chemiluminescent reactions, we elucidate the mechanism of chemiluminescence based on luminol and $H_2O_2$ with HRP as the catalyst, which is known as[17,18]:

$$HRP + H_2O_2 \rightarrow HRP-I + H_2O \tag{1}$$

$$HRP-I + LH^- \rightarrow HRP-II + L^{\cdot-} \tag{2}$$

$$HRP-II + LH^- \rightarrow HRP + L^{\cdot-} \tag{3}$$

$$L^{\cdot-} \rightarrow AP^{2-} + h\upsilon \tag{4}$$

HRP is first oxidized to HRP-I (one of the peroxidase reactive intermediates) by $H_2O_2$, while this product reacts with two molecules of luminol anions sequentially to produce two luminol ionic radicals ($L^{\cdot-}$), which are further oxidized to the excited state of 3-aminophthalic acid ($[AP^{2-}]^*$) leading to chemiluminescence[19], as shown in Fig. 1c. A detailed structure and explanation of these abbreviations is shown in Supplementary Table 1. 4-iodophenol can be introduced as an enhancer to increase the chemiluminescence intensity. 4-iodophenol's phenoxyl radicals can oxidize luminol and act as a redox mediator role; on the other hand, 4-iodophenol can react rapidly with the peroxidase reactive intermediates (HRP-I and HRP-II), thus accelerating the enzyme turnover frequency[17,20,21]. To image single-molecule chemiluminescence reactions and study the chemiluminescence kinetics, we added free HRP to the reaction system which is consistent with solution environments in the bulk assays[14,15,17], and elevated the focal plane into solution (rather than the surface of the glass coverslip) for detection. The emitted photons are then collected through the optical lens and detected by an electron-multiplying charge-coupled device (EMCCD) camera that enables high signal-noise ratio (SNR) single-photon detection.

To visualize a single event, we have employed a single-molecule spatiotemporal isolation strategy that controls the molecular density as well as the acquisition conditions for separating individual reactions in both space and time[1]. As chemiluminescence does not require laser excitation, the system background can be minimized to the detector offset level (Fig. 1d, e), which greatly facilitates the single-molecule observation with a high SNR (≈5). Figure 1f, g illustrates a typical on-off turnover chemiluminescence trajectory from a single pixel, a classic signature of single-molecule signals from stochastic reactions. The discrete number of acquired photons (1, 2, 3, ...) converted from the camera analog to digital (A/D) counts further indicates the observation of single-molecule reactions as a single photon is the output of an individual reaction. Thus, we managed to directly probe general single-molecule chemiluminescent reactions in solution, and our single-molecule chemiluminescence method can provide advantages with high spatiotemporal resolution compared with the bulk assays[14,15].

### Single-molecule analysis of HRP-catalyzed reactions

To understand the statistical nature of single-molecule reactions, we performed imaging under different exposure times and

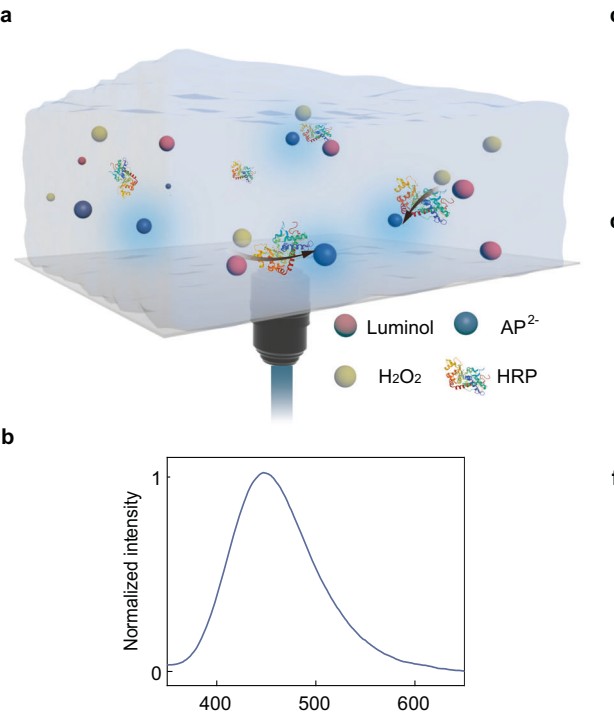

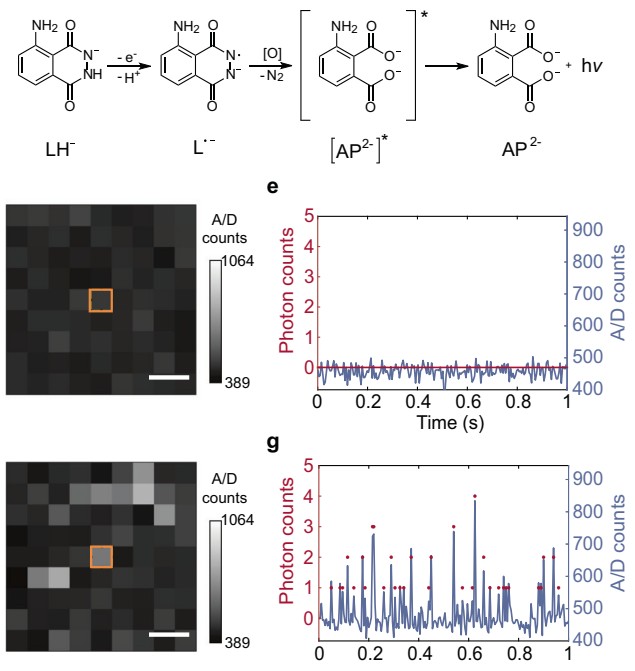

**Fig. 1 | Observation of single-molecule chemiluminescent reactions.**
**a** Schematic of chemiluminescence imaging system. **b** Chemiluminescence emission spectrum of luminol-$H_2O_2$ system. **c** Mechanism of luminol oxidation and chemiluminescence emission. [O] is the reactive oxygen species for $H_2O_2$ decomposition. **d** Image mode of background. Scale bar: 200 nm. **e** Camera A/D counts and photon counting trace of background. **f** Image mode of single-molecule chemiluminescence signals. Scale bar: 200 nm. **g** A/D counts and photon counting trace of single-molecule chemiluminescence signals. The chemiluminescence signals are observed at 1 mM luminol, 15 mM $H_2O_2$, 0.23 mM 4-iodophenol, 5 nM HRP with an exposure time of 5 ms and an electron-multiplying gain (EM Gain) of 500. Source data are provided as a Source Data file.

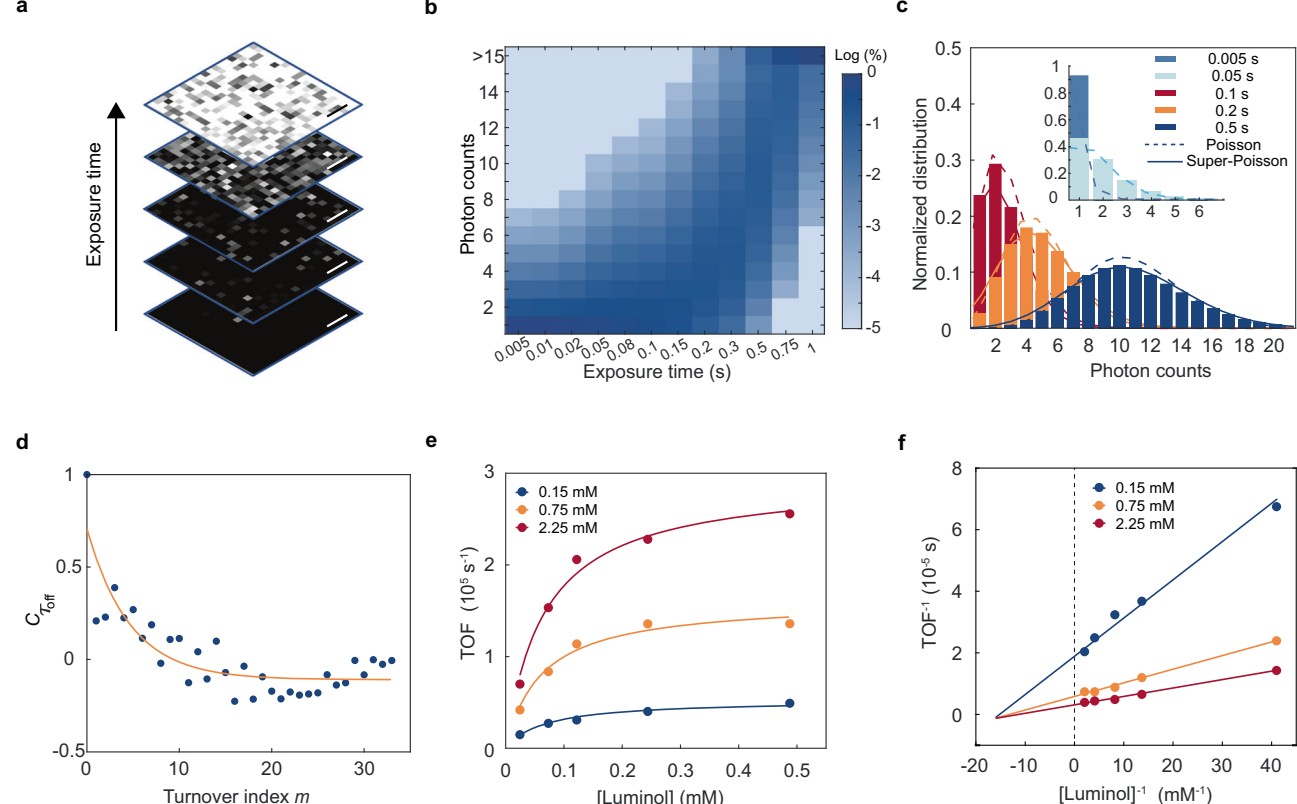

**Fig. 2 | Single-molecule analysis of HRP-catalyzed chemiluminescent reactions. a** Images at increasing exposure times. Scale bar: 400 nm. **b** Statistics of photon counts per pixel for different exposure times. **c** Poisson and super-Poisson distribution of photon counts. Lambda is defined here as the variance and expectation of the distribution. The lambda value is 0.09, 0.95, 1.71, 4.12, 9.84 under the exposure time of 0.005 s, 0.05 s, 0.1 s, 0.2 s, 0.5 s, respectively. **d** Autocorrelation function $C_\tau$ of $\tau_{off}$ from a single-molecule turnover trajectory. The $x$ axis is the turnover index. The yellow solid line is the fit with a single exponential for decay constant $m_{off}$ of 0.214 detected turnovers. **e** TOF as a function of luminol

concentration and the Michaelis-Menten equation fitting under three $H_2O_2$ concentrations. **f** Single-molecule Lineweaver-Burk plot corresponding to (**e**). The three lines intersect at a point to the left of the dashed line (abscissa <0). The chemiluminescence signal for different exposure times is acquired under 2.5 mM luminol, 15 mM $H_2O_2$, 0.23 mM 4-iodophenol, 5 nM HRP with EM Gain of 500. The chemiluminescence signal for different concentrations is acquired under 0.23 mM 4-iodophenol, 2.5 nM HRP with an exposure time of 5 ms and an EM Gain of 500. Source data are provided as a Source Data file.

analyzed the obtained photon number distribution in single pixels (Fig. 2a, b). The interval time between two events is defined as $\tau_{off}$, which is the time for product formation and contains abundant kinetic information of the chemiluminescent reactions. With the increase of exposure time, the number of accumulated photons increases (Supplementary Fig. 1, 2). As shown in Fig. 2c and Supplementary Fig. 3, photon number distribution shows a stochastic nature of chemiluminescent reactions at different exposure times, which initially follows a Poisson distribution and then transits to a super-Poisson distribution, indicating the reaction activity fluctuations.

The activity fluctuations are usually reflected by the reaction rate of the product formation process. We extracted $\tau_{off}$ from the single-molecule turnover trajectories and calculated the autocorrelation function[22], which displays an evident exponential decay with a decay constant of 0.214 detected turnovers, and further suggests the existence of activity fluctuations[22] (Fig. 2d).

From the stochastic single-molecule reaction trajectory, kinetic information of reactions can be obtained from the $\tau_{off}$ between two adjacent events (Supplementary Fig. 4, 5) or the turnover frequency (TOF), the number of photon counts per unit time[23],

$$TOF = \frac{\varphi_r \varphi_q \varphi_c \varphi_d N_m}{t} = \frac{N_d}{t} \qquad (5)$$

where $\varphi_r$ is the efficiency of the reaction, $\varphi_q$ is the luminescence quantum yield (determined by the luminescent substrate), $\varphi_c$ is the photon collection efficiency, $\varphi_d$ is the photoelectric conversion efficiency of EMCCD, $N_m$ is the number of molecules in collision, $N_d$ is the total number of detected photons, and $t$ is the acquisition time. The total efficiency in chemiluminescence is limited as all above efficiencies need to be considered. Based on $\tau_{off}$ and TOF analysis, we could explore the dynamics of single-molecule chemiluminescent reaction. In the presence of enhancer 4-iodophenol, we fixed the concentration of hydrogen peroxide (at 0.15, 0.75, and 2.25 mM) to figure out the luminol concentration dependence of the reaction kinetics. As shown in Fig. 2e, the single-molecule reaction kinetics evaluated under the HRP catalysis suggests an increasing TOF that tends to be stable with the increase of luminol concentration. Similarly, with a fixed concentration of luminol, the corresponding TOF also increases gradually with the growing $H_2O_2$ concentration. The obtained Lineweaver-Burk fitting result at the single-molecule level conforms to the characteristics of the double-substrate 'sequential' mechanism of Michaelis-Menten equation[24,25] (Fig. 2f),

$$TOF = \frac{V_{max}[H][L]}{K_m^H[L] + K_m^L[H] + K_{iH}K_m^L + [H][L]} \qquad (6)$$

where the $V_{max}$ is the maximum reaction rate, $K_m$ is the Michaelis-Menten constant, [H] and [L] are the concentration of $H_2O_2$ and

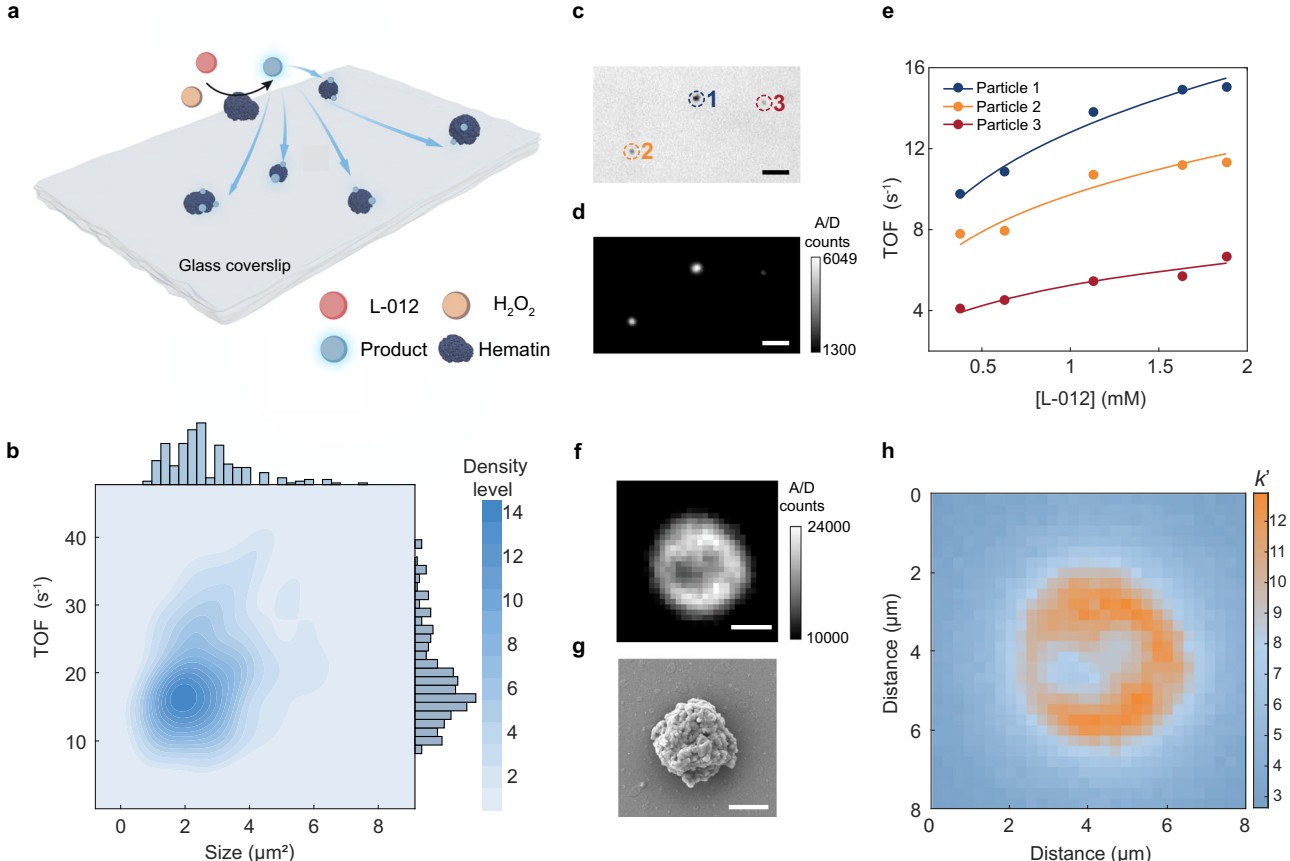

**Fig. 3 | Kinetics of hematin-catalyzed single-molecule chemiluminescent reactions. a** Schematic of hematin-catalyzed chemiluminescence imaging system. **b** Statistics of catalytic photon counts for over 200 hematin particles. **c** Bright field image and (**d**) chemiluminescence image of three different particles. Scale bar: 6 μm. **e** TOF as a function of L-012 concentration and kinetic curves fitting for three different hematin particles of (**c**). **f** Chemiluminescence image and (**g**) SEM image of a single hematin particle. Scale bar: 2 μm. **h** Transformed reaction rate constant ($k'$) mapping per pixel of corresponding sites of (**f**). Source data are provided as a Source Data file.

luminol, $K_{iH}$ is the dissociation constant of $H_2O_2$ and the enzyme HRP. The 'sequential' reaction mechanism scheme of double-substrate Michaelis-Menten equation is discussed in Supplementary Fig. 6. With this model, $K_m$ of luminol at different $H_2O_2$ concentrations was obtained (Supplementary Table 2). Similar results of the reaction kinetics were obtained from $\tau_{off}$ analysis (Supplementary Fig. 7) and power spectral density (PSD) analysis (Supplementary Fig. 8).

To further probe the reaction itself, we removed the enhancer. The TOF also increases with increasing concentration of luminol and $H_2O_2$, similar to the case with enhancer (Supplementary Fig. 9). However, the Lineweaver-Burk fitting shows a parallel relationship, suggesting a 'ping-pong' mechanism for the reaction of luminol and $H_2O_2$[24,26] (Supplementary Table 3). Moreover, chemiluminescence intensity without enhancer is 1 or 2 orders of magnitude lower than that with the presence of enhancer. Based on the experimental results and the role of 4-iodophenol for oxidating luminol[17], we propose a possible route leading to the mechanism shift (Supplementary Fig. 10).

### Single-molecule reaction kinetics on hematin particles
Using this approach, we are able to observe single-molecule chemiluminescent reactions in solution with freely diffusing HRP molecules or on single HRP particles (Supplementary Fig. 11). However, we did not manage to observe single-molecule chemiluminescent reaction

emission on single HRP molecules immobilized on a glass coverslip which is limited by the efficiency of the current chemiluminescence system (see results and discussions in Supplementary Fig. 12). The structure of HRP is often affected by the environment of chemiluminescence such as pH, ionic strength and temperature[27], which limits its catalytic activity and operations. Hematin, as the active center of HRP, features good stability and activity for catalyzing chemiluminescent reactions[28,29]. We immobilized hematin porcine catalyst particles on glass and performed chemiluminescence imaging for kinetics analysis. L-012 (8-amino-5-chloro-7-phenylpyrido[3,4-d]pyridazine-1,4(2H,3H) dione), which is the analogue of luminol with higher luminescence efficiency, is employed here as the luminescence substrate. L-012 reacts with $H_2O_2$ under the catalysis of hematin and generates light at 480 nm in this process (Fig. 3a and Supplementary Fig. 13). The chemiluminescence generated on hematin particles features a good stability, facilitating control experiments and correlative imaging (Supplementary Fig. 14). Compared with bright field, scanning electron microscopy (SEM), atomic force microscopy (AFM) imaging and energy dispersive spectrometer (EDS) mapping, an obvious diffusion can be seen in chemiluminescence images (Supplementary Figs. 15, 16), which is attributed to the long lifetime of the excited state of luminol and $H_2O_2$ intermediate radical[30]. According to the distribution of chemiluminescence, diffusion distance can be determined from imaging, and then lifetime of corresponding radicals (12.5 ms for $H_2O_2$ intermediate's radical and 18.9 ms for L-012 intermediate's radical) can be obtained

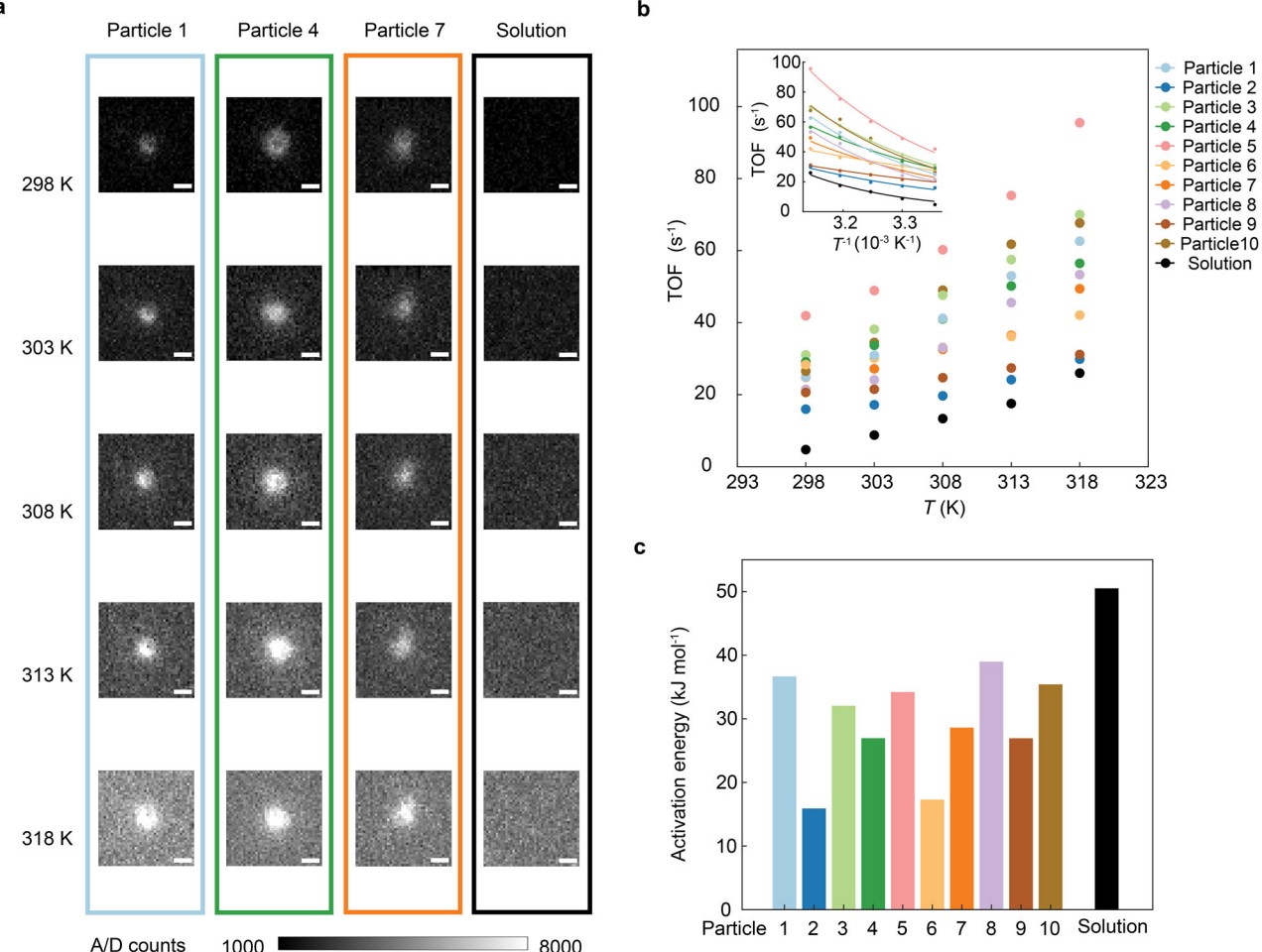

**Fig. 4 | Thermodynamics of hematin-catalyzed single-molecule chemiluminescent reactions. a** Chemiluminescence images of selected three hematin particles from (**b**) and solution at five different temperatures. Scale bar: 2 µm. **b** TOF on ten particles and in solution at different temperatures and corresponding Arrhenius equation fitting (the insert figure). **c** Activation energy extracted from (**b**). Source data are provided as a Source Data file.

(Supplementary Fig. 17). The SEM image shows that the catalyst particles are agglomerated by many strips of fine crystals, forming different shapes and sizes. On single catalyst particles, the distribution of photon number with the change of exposure time also conforms to a Poisson distribution (Supplementary Fig. 18). Statistical analysis of the size and TOF of all imaged hematin particles are shown in Fig. 3b. Based on the spatially resolved imaging, we can target hematin particles of interest for analysis of reaction kinetics (Fig. 3c–e):

$$TOF = \varphi k[L]^m[H]^n = k'[L]^m \qquad (7)$$

where $\varphi$ is the total efficiency, $k$ is the reaction rate constant, [H] and [L] are the concentration of L-012 and $H_2O_2$, m/n is the index item, and $k'$ is the transformed reaction rate constant. When the $H_2O_2$ concentration is fixed, the kinetics on hematin follows the Eq. (7), and the spatially resolved kinetics can be used to evaluate the heterogeneity across different catalyst particles.

Moreover, we selected a single catalyst particle for obtaining their chemiluminescence kinetics at different concentrations of L-012 (Fig. 3f–h, Supplementary Fig. 19). As shown in Fig. 3f, g, the single particle shows different catalytic behavior at different sites. Single-molecule kinetics at sub-particle sites follows the kinetics of Eq. (7) with the increasing L-012 concentration, but reveals different chemiluminescent reaction rate constant $k'$ on diverse sites, which reflects the catalytic heterogeneity at the sub-particle level (Fig. 3h).

### Single-molecule reaction thermodynamics on hematin particles

The ability to reduce the activation energy of the reaction system is a crucial indicator in evaluating the catalyst. We performed temperature-dependent experiments of the chemiluminescence reaction with hematin to explore the thermodynamics of this reaction (Supplementary Fig. 20). Figure 4a shows the chemiluminescence images under different temperatures of the solution and three representative catalyst particles (selected from ten particles in Fig. 4b). The thermodynamics of ten particles is obtained in Fig. 4b. As the temperature increases, TOF within the catalyst-free solution area increases rapidly with temperature while TOF of ten particles increases with a slower trend. Based on the TOF-temperature relationship, the activation energy can be estimated from the Arrhenius relationship (the insert figure in Fig. 4b)[31],

$$TOF = A'e^{\frac{-E_a}{RT}} \qquad (8)$$

where $E_a$ is the activation energy, $T$ is the temperature, R is the molar gas constant, and $A'$ is the transformed coefficient (see Methods). The calculated activation energy of the reaction in solution is 50.5 kJ mol$^{-1}$, and of these ten hematin particles is in the range of 10–40 kJ mol$^{-1}$ (Fig. 4c), which is in agreement with the activation energy range reported in the literature[32].

Our approach probes single-molecule chemiluminescent reaction kinetics in solution with freely diffusing HRP or on single catalyst particles with a single-photon sensitivity down to single-molecule reaction level, and spatially evaluates the thermodynamics of different particles, which can be used for investigating the relationships between the structures and the catalytic activity of catalysts. This method directly probes the light output of the chemical reaction, which minimizes the measurement background and the interference to the reaction system. The information obtained by the single-molecule chemiluminescence approach provides fundamental insights into the heterogeneity of both the kinetics and thermodynamics of catalytic reactions on single-particle level, offering a single-molecule methodology for studying chemical processes.

## Methods

### Single-molecule chemiluminescence measurement setup
Single-molecule chemiluminescence measurements were performed on an inverted optical microscope (IX83, Olympus). Chemiluminescence generated by the reactants was collected by a 150× oil objective (Olympus, numerical aperture: 1.45) for HRP-catalyzed experiments or a 60× oil objective (Olympus, numerical aperture: 1.49) for hematin porcine-catalyzed experiments, respectively, and detected by an EMCCD camera (iXon Ultra 897, Andor) with an EM Gain of 500.

### Detection of single-molecule chemiluminescence in solution
The glass coverslip used for holding aqueous sample solution was sonicated in acetone, isopropanol, and deionized water sequentially for 30 min, followed by nitrogen drying, and then fixed into the microscope sample chamber. 2 mL-volume mixture of Tris-HCl buffer (0.1 M, pH = 8.5), HRP (ThermoFisher Scientific), luminol ($C_8H_7N_3O_2$, Aladdin, 98%) and 4-iodophenol ($C_6H_5IO$, Macklin, 98%) solution was added to the sample chamber, and $H_2O_2$ solution was added to the whole solution which was then prepared under stirring at room temperature for further experiments.

### Chemiluminescence imaging of hematin porcine particles
First, hematin porcine ($C_{34}H_{33}N_4O_5Fe$, Sigma, 96%) powder was dispersed in ethanol and treated with a 20-min-long ultrasound process for homogenizing, then 100 μL hematin solution was added onto the glass coverslip and kept for 20 min until the ethanol evaporated completely, and the unbound particles could be rinsed with deionized water. The glass coverslip was then fixed into the PTFE sample chamber with a gasket, and 2 mL mixture of Tris-HCl buffer and L-012 solution ($C_{13}H_9ClN_4O_2$, Fujifilm, 90%) was added, followed by adding $H_2O_2$ solution and fully stirring to start the reaction. Detection was performed after the signal stabilizes for 1–2 min.

### Temperature-changing experiment for hematin porcine particles
Hematin porcine particles were deposited on glass coverslip. The polyimide heating film was wrapped on the outer surface of the sample chamber, and was connected to the temperature control system. After adding buffer and substrate solution, the temperature of the solution was heated to the set value via the controller, afterwards the chemiluminescence was detected by an EMCCD.

### Correlation of characterization methods
To carry out the correlation experiments, an array of Cr-Au was fabricated on the glass coverslip for navigation. Then the bright field and chemiluminescence images were taken. In addition, SEM (ZEISS Gemini 300) and AFM (Molecular Vista VistaScope) images were acquired to obtain the morphology and structure of catalyst particles. EDS mapping was used to determine the composition of the catalyst.

### Autocorrelation analysis
We analyzed the autocorrelation using following equation,

$$C_\tau(m) = <\Delta\tau(0)\Delta\tau(m)>/<\Delta\tau^2> \tag{9}$$

where m is the turnover index of the sequence, and $C_\tau(m)$ is the autocorrelation function.

### Transformation of the Arrhenius equation

$$TOF = \varphi[L]^m[H]^n \times k = \varphi[L]^m[H]^n \times Ae^{\frac{-E_a}{RT}} = A'e^{\frac{-E_a}{RT}} \tag{10}$$

with $E_a$: activation energy, $A$: pre-exponential factor, $T$: temperature, R: molar gas constant, $A'$: the transformed coefficient.

### Reporting summary
Further information on research design is available in the Nature Portfolio Reporting Summary linked to this article.

## Data availability
The data that support the findings of this study are available from the corresponding authors upon request. Source data are provided with this paper.

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

## Acknowledgements

This study was supported by the Fundamental Research Funds for the Central Universities (grant number: K20220088), the National Natural Science Foundation of China (grant number: 21974123), the National Key R&D Program of China (grant number: 2020YFA0211200) and the Natural Science Foundation of Zhejiang Province (grant number: LR20B050002) to J.F. We thank the Micro and Nano Fabrication Center and the Chemistry Instrumentation Center at Zhejiang University for facility support.

## Author contributions

Z.Z. performed the experiments, analyzed the data and wrote the manuscript; J.D. helped with the experiments and analyzed the data; Y.Y. analyzed the data and performed fluorescence experiments; Y.Z. and Y.C. performed the SEM characterizations; Y.X. performed the AFM characterizations; J.F. conceived the project, supervised the study and wrote the manuscript.

## Competing interests

The authors declare no competing interests.
