## [Peer Review File · Nature Communications]

Direct probing of single-molecule chemiluminescent reaction dynamics under catalytic conditions in solutionREVIEWER COMMENTS

Reviewer #1 (Remarks to the Author):

The authors previously reported on single molecule observations of electrochemiluminescence. In the present report, the authors applied their experimental setup to observe the dynamics of chemiluminescence reactions at the single molecule level. This was accomplished without the need for laser excitation, which causes large backgrounds and autofluorescence. This is the first report that deals with the dynamics of chemiluminescence at the single molecule level. This is a nice paper and is recommended for publication in Nature Commun, provided that the authors address the following points before publication

1. equations (1)-(5): Explain abbreviations such as LH- and AP. Use these abbreviations in Figures 1 A and B as well.
2. Explain what the exact product is in Figures 1A and 3A.
3. The two main parameters discussed in this manuscript, t_{OFF} and ϕ , are both affected by the quantum yield of emission. The heterogeneity observed in this manuscript may reflect heterogeneity in luminescence quantum yields rather than catalytic activity (another interesting phenomenon).
4. Diffusion can be seen in the chemiluminescence image. Using the diffusion coefficient of water, it would be possible to determine the lifetime of the excited state from this image. It may also be possible to discuss the luminescence quantum yield from this data.
5. It is recommended that a brief description of the "ping-pong" mechanism be provided.

Reviewer #2 (Remarks to the Author):

In this manuscript, Zhang et al. measure chemiluminescence reactions on the single-molecule level. To this end, they observe the reaction of luminol with H_2O_2 catalyzed by HRP proteins via high-speed microscopy. They extract reaction rates and analyze them via Michaelis Menten kinetics. They then apply a similar workflow to study luminol chemoluminescence catalyzed by hematin particles, where they observe non-homogeneous catalysis across the catalyst particles and where they determine apparent activation energies between different particles.

The ability to observe chemical reactions on the single-molecule level can be very powerful and chemoluminescent reactions are very interesting. The authors use powerful imaging and observe the reactions with single-photon detection. However, in my opinion the study requires some further characterization of the system, as described below.

1. Regarding this being described as a single-molecule assay: Under each pixel, there are probably many HRP molecules. Moreover, the concentrations of Luminol and H_2O_2 are very high (mM). The authors use a high framerate of detection, so only few photons per pixel are detected (with unclear detection efficiency). But I fail to see the difference to a bulk assay. The authors should better specify what the added value is here. E.g. if we would be able to observe the behavior of individual catalysts (i.e. turnover etc.) we would gain information about HRP, or the molecular role of 4-iodophenol. But such as described here, I am not sure what the goal is.
2. The authors should specify their experimental procedure.
 - Are they immobilizing HRP on the glass surface?
 - What is the concentration of the protein, and how densely is the surface populated by protein molecules?
 - How stably do the HRP molecules stick to the glass?
 - Could the observed fluctuation in activity arise from diffusing HRP, or other glass-interface dependent mechanisms?

Given that they add the protein to non-passivated glass, this is most likely, albeit the sketch in Fig. 1a implies that the protein is freely diffusing. Of note, most protein-based single-molecule experimental systems avoid this by passivating the glass surface, e.g. via a PEG layer, avoiding nonspecific

interactions. Then, proteins can be immobilized in a controlled fashion, e.g. via biotin-neutravidin attachment chemistry.

3. What are the imaging conditions, and what is the pixel size shown in Fig. 1 and 2?

4. The authors should discuss the role of 4-iodophenol in the process

In the reactions (1) – (4) they should specify the identity of the reagents, i.e. LH-, L-, HRP, HRP-I, HRP-II or AP2- (also luminol, I guess). What is the difference between luminol free radicals and L.-?

Figure 1B: In this reaction equation, the role of H₂O₂ and the N₂ loss should be added (I don't know what [O] is)

5. It is interesting that the authors observe a change in mechanism dependent on the presence of 4-iodophenol. The authors do however not comment on why this is. Also they should indicate how their 'sequential' and 'ping-pong' mechanism map on the chemical reactions described in eq. 1-4. Indeed a sequential mechanism does not seem to make sense, as the oxidation of HRP seems to precede the subsequent steps.

6. The temperature dependence of chemoluminescence seems to show a systematic deviation from the Arrhenius fit, i.e. an underestimated curvature. What is the reason for this? Also, the E_a measurements are lacking statistics. Are individual particles indeed performing differently as catalysts, or is this just within the error of the experiment?

Response to reviewer comments

Dear Editor and Reviewers:

Thank you all for evaluating our manuscript. We are grateful for the high-quality comments from our reviewers, which helped us improve our paper. Detailed responses to all reviewer comments can be found in this letter and revisions in the manuscript are highlighted in red text.

Reviewer 1: The authors previously reported on single molecule observations of electrochemiluminescence. In the present report, the authors applied their experimental setup to observe the dynamics of chemiluminescence reactions at the single molecule level. This was accomplished without the need for laser excitation, which causes large backgrounds and autofluorescence. This is the first report that deals with the dynamics of chemiluminescence at the single molecule level. This is a nice paper and is recommended for publication in Nature Commun, provided that the authors address the following points before publication.

We thank the reviewer for summarizing the main contributions of our work and for the strong recommendation as well as the constructive comments.

(1.1). equations (1)-(5): Explain abbreviations such as LH⁻ and AP. Use these abbreviations in Figures 1 A and B as well.

Response: We have added **Table R1** to explain the abbreviations as blow (which is also included as **Supplementary Table. 1**). These abbreviations are now also explained in **Fig. 1A**, **Fig. 1B** and page 4 of the revised manuscript.

Table R1. Abbreviations and corresponding explanations.

Abbreviation	Name	Structural formula
LH ⁻	luminol (5-amino-2,3-dihydro-1,4-phthalazinedione)	L ⁻	luminol ionic radical	AP ²⁻	3-aminophthalic acid	HRP	horseradish peroxidase	PDB DOI for HRP: https://doi.org/10.2210/pdb1H58/pdb
HRP-I, HRP-II	peroxidase reactive intermediates	PDB DOI for HRP-I: https://doi.org/10.2210/pdb1HCH/pdb

		PDB DOI for HRP-II: https://doi.org/10.2210/pdb1H55/pdb
[O]	reactive oxygen species for H ₂ O ₂ decomposition	HO ₂ ⁻ / O ₂ ⁻
L-012	8-amino-5-chloro-7-phenylpyrido[3,4-d]pyridazine-1,4(2H,3H)dione	
Product of oxidized L-012	8-amino-5-chloro-7-phenylpyrido[3,4-d]pyridazine-1,4(2H,3H)dicarboxylic acid	

We have modified **Fig. 1A** and **Fig. 1B** in the manuscript:

(1.2). Explain what the exact product is in Figures 1A and 3A.

Response: In **Fig. 1A**, the exact product 3-aminophthalic acid is produced by the oxidation of

luminol, and its specific structural formula is . Similarly, in **Fig. 3A**, the exact product 8-amino-5-chloro-7-phenylpyrido[3,4-d]pyridazine-1,4(2H,3H)dicarboxylic acid is

produced by the oxidation of L-012, and its specific structural formula is

(1.3). The two main parameters discussed in this manuscript, tOFF and phi, are both affected

by the quantum yield of emission. The heterogeneity observed in this manuscript may reflect heterogeneity in luminescence quantum yields rather than catalytic activity (another interesting phenomenon).

Response: Luminescence quantum yield of chemiluminescence is defined as the photon emission probability of a single substrate molecule in the reaction¹. In our experiments, when calculating TOF, we consider that the quantum yield of emission in chemiluminescence is determined by the luminescent substrate itself (luminol and L-012). Numerous studies in the literature have shown that the quantum yield of luminol chemiluminescence in aqueous solution is 1.1-1.2%, and the quantum yield is considered to be constant for a wide range of luminol concentrations¹⁻³. L-012's quantum yield of emission is higher than that of luminol. Because the substrate species and the solution environment remained unchanged in our experiment, we consider that the quantum yield of emission in chemiluminescence is constant. To explain the effect of luminescence quantum yields, we add a discussion in the page 7 of the manuscript,

“

$$TOF = \frac{\varphi_r \varphi_q \varphi_c \varphi_d N_m}{t} = \frac{N_d}{t} \quad (5)$$

where φ_r is the efficiency of the reaction, φ_q is the luminescence quantum yield (determined by the luminescent substrate), φ_c is the photon collection efficiency, φ_d is the photoelectric conversion efficiency of EMCCD, N_m is the number of molecules in collision, N_d is the total number of detected photons, and t is the acquisition time (s). The total efficiency in chemiluminescence is limited as all above efficiencies need to be considered.”

Meanwhile, hematin is a known catalyst to accelerate the chemiluminescence reaction by decreasing the activation energy (influencing φ_r) and increasing the number of luminol molecules in the reaction, so we believe that the heterogeneity of the hematin particles originates from the difference in catalytic activity rather than luminescence quantum yields.

(1.4). Diffusion can be seen in the chemiluminescence image. Using the diffusion coefficient of water, it would be possible to determine the lifetime of the excited state from this image. It may also be possible to discuss the luminescence quantum yield from this data.

Response: We thank the reviewer for this important suggestion. In the revised version, we have added a brief discussion of diffusion distance and intermediate lifetime in page 9 of manuscript and a detailed description in Supplementary information as **Supplementary Fig. 15**.

Revision in page 9 of manuscript,

“According to the distribution of chemiluminescence, diffusion distance can be determined from imaging, and then lifetime of corresponding radicals can be obtained (Supplementary Fig. 15).”

Revision in Supplementary information is shown in **Supplementary Fig. 15**,

Supplementary Fig. 15. Chemiluminescence intensity distribution and diffusion analysis on the hematin particle. (A) Bright field and (B) chemiluminescence image of a hematin particle. Scale bar: 5 μm . (C) Normalized intensity profiles along the radial direction of (A) bright field and (B) chemiluminescence. Diffusion distance is from catalyst surface (the position of the black dashed line, and the normalized intensity of bright field is 0) to the position where the normalized chemiluminescence intensity is 0.

“Obvious chemiluminescence diffusion was visible on the hematin catalysts. **Supplementary Fig. 15A and B** show the bright field and chemiluminescence image of the same hematin particle. Normalized intensity profiles along the radial direction are shown in **Supplementary Fig. 15C**. According to the distribution, chemiluminescence intensity decreases more slowly than that of the bright field with the direction away from the catalyst surface. The distance from the catalyst surface (the position that the normalized intensity of bright field is 0) to the position where the chemiluminescence intensity is 0 is considered to be the diffusion distance of radicals’ intermediates. As shown in **Supplementary Fig. 15C**, the diffusion distance on the catalyst particles is $\sim 5 \mu\text{m}$, which is comparative to the previous result in the literature⁴. According to the equation,

$$L = \sqrt{2D\tau}$$

where L is the diffusion distance, D is the diffusion coefficient of radicals, $D_{L-012} = 6.6 \times 10^{-6} \text{ cm}^2/\text{s}$, $D_{\text{H}_2\text{O}_2} = 1 \times 10^{-5} \text{ cm}^2/\text{s}$, and τ is the radical lifetime⁴⁻⁶, the H_2O_2 intermediate’s radical lifetime is estimated to be 12.5 ms and L-012 intermediate’s radical lifetime is 18.9 ms.”

(1.5). It is recommended that a brief description of the “ping-pong” mechanism be provided.

Fig. R2. ‘Ping-pong’ double-substrate enzyme mechanism and the Lineweaver-Burk plot. (A) Double-substrate ‘ping-pong’ mechanism. (B) Michaelis-Menten equation fitting of ‘ping-pong’ mechanism at different luminol and H₂O₂ concentrations. (C) Single-molecule Lineweaver-Burk plot.

Response: As shown in **Fig. R2A (Supplementary Fig. 9A)**, substrate (A or B) and corresponding product (P or Q) are alternately bound to or released from the enzyme (E), one after the other, just like in a game of ping-pong, so it is called the ‘ping-pong’ mechanism^{4,5}. The ping-pong mechanism is described in **Supplementary Fig. 9**,

“the first substrate A combines with enzyme E to form EA, then the functional group of A is transferred to E to form E’P, which releases P to form E’. E’ combines with the second substrate B following the same principles to form Q and release E. In the whole reaction process, only the binary complex form exists, and the ternary complex form is absent. The ‘ping-pong’ mechanism follows,

$$TOF^{-1} = \frac{1}{V} = \frac{K_m^H}{V_{max}[H]} + \frac{K_m^L}{V_{max}[L]} + \frac{1}{V_{max}}$$

„

The kinetic equation for the ‘ping-pong’ mechanism features multiple parallel lines, as given by the results in **Fig. R2 B** and **C**.

Reviewer 2: In this manuscript, Zhang et al. measure chemiluminescence reactions on the single-molecule level. To this end, they observe the reaction of luminol with H₂O₂ catalyzed by HRP proteins via high-speed microscopy. They extract reaction rates and analyze them via Michaelis Menten kinetics. They then apply a similar workflow to study luminol chemoluminescence catalyzed by hematin particles, where they observe non-homogeneous catalysis across the catalyst particles and where they determine apparent activation energies between different particles. The ability to observe chemical reactions on the single-molecule level can be very powerful and chemiluminescent reactions are very interesting. The authors use powerful imaging and observe the reactions with single-photon detection. However, in my opinion the study requires some further characterization of the system, as described below.

We thank the reviewer for the appreciation of our work and for highlighting the value of single molecule chemiluminescence reaction observations as well as our imaging power. In the revised manuscript, following the reviewer's comments and suggestions, we have made a through revision to improve the characterization of the system and the paper quality.

(2.1). Regarding this being described as a single-molecule assay: Under each pixel, there are probably many HRP molecules. Moreover, the concentrations of Luminol and H₂O₂ are very high (mM). The authors use a high framerate of detection, so only few photons per pixel are detected (with unclear detection efficiency). But I fail to see the difference to a bulk assay. The authors should better specify what the added value is here. E.g. if we would be able to observe the behavior of individual catalysts (i.e. turnover etc.) we would gain information about HRP, or the molecular role of 4-iodophenol. But such as described here, I am not sure what the goal is.

Response: We thank the reviewer for this critical comment. Since chemiluminescence is limited by the efficiency of several processes, we used mM substrate concentrations for chemiluminescence imaging and detection. And we add a discussion in the page 7 of the revised manuscript:

“

$$TOF = \frac{\varphi_r \varphi_q \varphi_c \varphi_d N_m}{t} = \frac{N_d}{t} \quad (5)$$

where φ_r is the efficiency of the reaction, φ_q is the luminescence quantum yield (determined by the luminescent substrate), φ_c is the photon collection efficiency, φ_d is the photoelectric conversion efficiency of EMCCD, N_m is the number of molecules in collision, N_d is the total number of detected photons, and t is the acquisition time (s). The total efficiency in chemiluminescence is limited as all above efficiencies need to be considered.”

Compared to bulk assay, our method provides more information with high spatiotemporal resolution. Firstly, in our experiments we used a high acquisition frequency (189.97 Hz) to obtain single-molecule chemiluminescence signals with temporal resolution (**Fig. 1D**), which improves the detection of chemiluminescence to a single photon level. Secondly, the imaging observation by a high NA (1.45) objective microscope and a highly sensitive EMCCD detector (quantum efficiency: 83% at 425 nm and 90% at 485 nm) offers high-throughput reaction

information with spatial resolution and high detection efficiency of the imaging system, whereas the spatial and temporal information is very limited in the previous bulk assays.

Indeed, in the first part of our manuscript we used free immobilized HRP enzyme molecules in solution, and not much spatial information was utilized. To further demonstrate the advantages of our method, following the suggestion from the reviewer, we physically deposited individual HRP particles on glass slides and selected two HRP catalysts for characterization, as shown in **Fig. R3A**. **Fig. R3B** shows single turnover signal of chemiluminescent reactions on individual HRP particles. Compared with camera background, single-molecule chemiluminescence on individual catalysts can be observed visually. In addition, we selected one portion of the sequence (black dashed box) for further analysis (**Fig. R3C**), and the results show that the catalytic behavior of HRP exhibits a fluctuation (clustered burst events) which is different from the bulk assay, reflecting the dynamic heterogeneity and fluctuation of the catalytic activity of individual catalysts and the advantage of high spatiotemporal resolution in our approach.

Fig. R3. Catalytic behavior on individual HRP particles. (A) Chemiluminescence of individual HRP particles on glass coverslip. Exposure time: 10 s, EM Gain= 500, 60× oil objective. Scale bar: 5 μm (B) A/D counts of two selected particles in (A) and camera background. Exposure time: 5 ms, EM Gain= 500. (C) Single-molecule chemiluminescence analysis. The data is from the selected area (black dashed box) of (B).

In order to highlight the advantages of single-molecule detection method in this paper, we added following sentence in the page 5 in the revised version:

“and our single-molecule chemiluminescence method can provide advantages with high spatiotemporal resolution compared with the bulk assays^{14, 15}.”

(2.2). The authors should specify their experimental procedure.

- Are they immobilizing HRP on the glass surface?
- What is the concentration of the protein, and how densely is the surface populated by protein molecules?
- How stably do the HRP molecules stick to the glass?
- Could the observed fluctuation in activity arise from diffusing HRP, or other glass-interface dependent mechanisms?

Given that they add the protein to non-passivated glass, this is most likely, albeit the sketch in Fig. 1a implies that the protein is freely diffusing. Of note, most protein-based single-molecule experimental systems avoid this by passivating the glass surface, e.g. via a PEG layer, avoiding nonspecific interactions. Then, proteins can be immobilized in a controlled fashion, e.g. via biotin-neutravidin attachment chemistry.

Response: Our single molecule study refers to imaging single-molecule chemiluminescence reactions. In our manuscript, in order to better align with the environments and enzyme states in previous chemiluminescence assays, so as to control conditions to further study the kinetics of chemiluminescence at the single-molecule level, we finally chose to conduct the experiment in the free-enzyme environment for this purpose without fixing the HRP on the glass surface for **Fig. 1** and **Fig. 2**. The used concentration of free HRP is 5 nM. During the detection of single-molecule chemiluminescence, the focal plane stayed over 10 μm above the glass coverslip to ensure that the observed HRP molecules were free in solution. In Fig. 2D, we discussed the activity fluctuations reflected by the detected photons from the single-molecule turnover trajectories. As only a limited number of HRP is available considering the pixel size (106.67 nm) and depth of field (101.1 nm), we hypothesize that the activity fluctuations originate from the free diffusion of HRP in solution.

In this revision, we have tried a number of approaches to immobilize single HRP molecules onto glass coverslip and tried to obtain the single-molecule chemiluminescence signals. We used the same method reported in H. Peter Lu and X. Sunney Xie's previous work⁶, going through a series of immobilizations using NaOH-ethanol solution, 10% TESPA and isobutyltrimethoxysilane (1:10,000 ratio) in DMSO solution, 10 nM DMS·2HCl and 1 nM HRP. In this way, HRP molecules are fixed to the surface of the glass coverslip at a density of ~ 0.2 molecule/ μm^2 . However, no obvious chemiluminescence signals can be seen from only a single HRP molecule immobilized on the glass coverslip, as shown in **Fig. R4A-C**. Besides, we also tried some other immobilization methods: such as NaOH-ethanol solution, 10% TESPA and isobutyltrimethoxysilane (1:10,000 ratio) in DMSO solution, 10 nM NHS-PEG (5000)-biotin, and 10 nM Streptavidin-HRP, and the results were similar.

In order to verify whether the HRP was indeed immobilized to the surface of the glass coverslip, we used amplex red, a fluorescence probe which can be catalyzed by HRP in the presence of H_2O_2 and emit fluorescence with an emission peak of 585 nm. **Fig. R4E** shows the fluorescence signal with obvious scintillation, and the position of the single HRP molecule is faintly visible. The 'on-off' trace shown in **Fig. R4D** is the typical single-molecule turnover trajectory. Therefore, the fluorescence experiment indicates that the single HRP molecule was successfully immobilized to the glass coverslip. The fluorescence images and the enzyme

density of HRP attached to the glass remained unchanged after several procedures of solution changing and washing, which shows that the HRP immobilization had good stability. However, the single-molecule chemiluminescence imaging did not provide any obvious signal. We suppose that HRP is a catalyst for both fluorescence and chemiluminescence, however, compared to fluorescence system, the luminescence efficiency of chemiluminescence is much lower (e.g. luminol's quantum yield, 1.1%-1.2%), resulting in the too weak single molecule chemiluminescence at the single enzyme molecule level to be detected at present. In the future, if high efficiency chemiluminescence substrates or additives can be developed, the single HRP molecule detection and kinetics analysis might be possible. We would like to clarify that our single molecule study refers to imaging single-molecule chemiluminescence reactions, focusing on the capability of observing individual chemiluminescence reactions rather than studying single enzyme molecules. We have modified the manuscript to avoid any confusion.

Fig. R4. Single HRP molecule detection in chemiluminescence and fluorescence on glass coverslip. (A) Intensity of camera background and chemiluminescence. Exposure time: 5 ms, EM Gain= 500. (B) Chemiluminescence image of the glass with immobilized HRP. Exposure time: 10 s, 100 \times oil objective, EM Gain= 500. Scale bar: 2 μ m. (C) TOF analysis of selected 100 \times 100 pixels in chemiluminescence of (B). (D) Intensity of fluorescence on a single HRP molecule. Exposure time: 20 ms. 100 \times oil objective. (E) Fluorescence image of single HRP molecule. Scale bar: 2 μ m.

To specify our experimental procedure, we added following sentence in the page 5 of the

revised version:

“In order to image single-molecule chemiluminescence reactions and study the chemiluminescence kinetics, we added free HRP to the reaction system which is consistent with solution environments in the bulk assays^{14, 15, 17}, and elevated the focal plane into solution (rather than the surface of the glass coverslip) for detection.”

(2.3). What are the imaging conditions, and what is the pixel size shown in Fig. 1 and 2?

Response: Single-molecule chemiluminescence measurements were performed on an inverted optical microscopy (IX83, Olympus). Chemiluminescence generated by the reactants in **Fig. 1** and **Fig. 2** was collected by a 150× oil objective (Olympus, numerical aperture: 1.45, pixel size: 106.67 nm) for HRP-catalyzed experiments and detected by an EMCCD camera (iXon Ultra 897, Andor) with an EM Gain of 500 and an exposure time of 5 ms. Imaging conditions are mentioned in Supplementary information and Figure captions.

(2.4). The authors should discuss the role of 4-iodophenol in the process

In the reactions (1) – (4) they should specify the identity of the reagents, i.e. LH-, L.-, HRP, HRP-I, HRP-II or AP2- (also luminol, I guess). What is the difference between luminol free radicals and L.-?

Figure 1B: In this reaction equation, the role of H₂O₂ and the N₂ loss should be added (I don't know what [O] is)

Response: For the role of 4-iodophenol in the process, we have added the revision in the page 4 of the manuscript:

“4-iodophenol can be introduced as an enhancer to increase the chemiluminescence intensity. 4-iodophenol's phenoxyl radicals can oxidize luminol and act as a redox mediator role; on the other hand, 4-iodophenol can react rapidly with the peroxidase reactive intermediates (HRP-I and HRP-II), thus accelerating the enzyme turnover frequency^{17, 20, 21}.”

For the reactions (1) – (4), we have added the **Table R1**. to define the abbreviations as blow. Besides, N₂ have been added in the **Fig. 1B**, and [O] is the reactive oxygen species for H₂O₂ decomposition.

The revised Fig. 1B is:

Table R1. Abbreviations and corresponding explanations.

Abbreviation	Name	Structural formula
--------------	------	--------------------

LH^-	luminol (5-amino-2,3-dihydro-1,4-phthalazinedione)	
$L^{\cdot-}$	luminol ionic radical	
AP^{2-}	3-aminophthalic acid	
HRP	horseradish peroxidase	PDB DOI for HRP: https://doi.org/10.2210/pdb1H58/pdb
HRP-I, HRP-II	peroxidase reactive intermediates	PDB DOI for HRP-I: https://doi.org/10.2210/pdb1HCH/pdb PDB DOI for HRP-II: https://doi.org/10.2210/pdb1H55/pdb
[O]	the reactive oxygen species for H_2O_2 decomposition	$HO_2^- / O_2^{\cdot-}$
L-012	8-amino-5-chloro-7-phenylpyrido[3,4-d]pyridazine-1,4(2H,3H)dione	
Product of oxidized L-012	8-amino-5-chloro-7-phenylpyrido[3,4-d]pyridazine-1,4(2H,3H)dicarboxylic acid	

(2.5). It is interesting that the authors observe a change in mechanism dependent on the presence of 4-iodophenol. The authors do however not comment on why this is. Also they should indicate how their ‘sequential’ and ‘ping-pong’ mechanism map on the chemical reactions described in eq. 1-4. Indeed a sequential mechanism does not seem to make sense, as the oxidation of HRP seems to precede the subsequent steps.

Response: We thank the reviewer for this comment. In our experiments, after the addition of 4-iodophenol, **Fig. 2E** and **Fig. 2F** reveal that the 3 curves for different H_2O_2 concentrations intersect at one point, a result that is consistent with the kinetics of the ‘sequential’ mechanism. For the shift in the mechanism, we added a detailed discussion in page 18 of the revised Supplementary information:

“For the ‘ping-pong’ mechanism with the absence of 4-iodophenol, the substrate H_2O_2 , binds to HRP to produce the [O] and HRP intermediates: HRP-I. And the subsequent substrate LH^- ,

then reacts with HRP-I and HRP-II to produce AP^{2-} and restore HRP to its initial form. However, in the 'sequential' mechanism (see Supplementary Fig. 6), substrates H_2O_2 and LH^- both have to bind to the HRP before releasing the products and restoring the HRP to its initial state, which is one reaction step less compared to the 'ping-pong' mechanism. In the first step, 4-iodophenol acts as a redox mediator to directly oxidize LH^- to $L^{•-}$, resulting in the early oxidation of luminol³, which advances and accelerates the processing of reaction and exhibits characteristics similar to the first step of the 'sequential' mechanism. Besides, 4-iodophenol's free radical intermediates then react with HRP-I to generate HRP-II and accelerate the enzyme turnover frequency so that more LH^- molecules are involved in the reaction at the same time. In combination with the above two steps of acceleration and the early involvement of LH^- in the reaction, the kinetics of the reaction shows a mechanism shift."

Supplementary Fig. 10 (Fig. R5) is also added in Supplementary information.

'ping-pong' mechanism in this work:

'sequential' mechanism in this work:

Figure R5. 'Ping-pong' and 'sequential' reaction mechanism scheme in this work. The direction of the arrow is reflective of the direction of the reaction process.

(2.6). The temperature dependence of chemiluminescence seems to show a systematic derivation from the Arrhenius fit, i.e. an underestimated curvature. What is the reason for this? Also, the E_a measurements are lacking statistics. Are individual particles indeed performing differently as catalysts, or is this just within the error of the experiment?

Response: We thank the reviewer for raising this point. The catalyst stability can be susceptible to a large effect when the temperature was too high due to the set temperature interval being too large (up to 333 K). Besides, the low statistics of the previous particles and the neglect of the noise photons from camera background also led to the large deviation of the data, resulting in a poor fit and an insufficient representation of the catalytic behavior on hematin shown in our previous data. To address this point, we reset the temperature interval (298-318 K), selected improved number of particles (improved from 2 to 10) for observation and subtracted the noise photons of the camera background during data processing. The results are shown in **Fig. R6 (Fig. 4)**. The activation energies on the catalysts are all lower than that in solution (50.5 kJ/mol), with a particle-to-particle heterogeneity (10~40 kJ/mol). To clarify this difference is not from

the experiment error, we measured single particles at difference times and the chemiluminescence intensity remains stable (RSD = 5%) and does not produce large error that can confuse the observed particle-to-particle heterogeneity (Fig. R7). We thus conclude that our experiments show the different catalytic effects of individual particles.

Fig. R6 (Fig. 4). Thermodynamics of hematin-catalyzed single-molecule chemiluminescent reactions. (A) Chemiluminescence images of selected three hematin particles from (B) and solution at five different temperatures. Scale bar: 2 μm . (B) TOF on ten particles and in solution at different temperatures and corresponding Arrhenius equation fitting (the insert figure). (C) Activation energy extracted from (B).

Fig. R7. Chemiluminescence stability of the hematin particle. Exposure time: 10 s, EM Gain: 500.

In the revised version, we have added corresponding description of Fig. 4 in page 10:

“Chemiluminescence images under different temperatures of the solution and three representative catalyst particles (selected from ten particles in Fig. 4B) are shown in Fig. 4A. And the thermodynamics of ten particles is obtained in Fig. 4B. As the temperature increases, TOF within the catalyst-free solution area increases rapidly with temperature while TOF of ten particles increases with a slower trend (Fig. 4B).”

and another description in page 10:

“The calculated activation energy of the reaction in solution is 50.5 kJ/mol, and of these ten hematin particles is in the range of 10~40 kJ/mol (Fig. 4C).”

To describe the stability of chemiluminescence at different temperatures, we add **Supplementary Fig. 18 (Fig. R7)** in revised Supplementary information.

References:

1. Ando, Y. et al. Development of a quantitative bio/chemiluminescence spectrometer determining quantum yields: re-examination of the aqueous luminol chemiluminescence standard. *Photochem. Photobiol.* **83**, 1205-1210 (2007).
2. Lee, J. & Seliger, H.H. Absolute spectral sensitivity of phototubes and the application to the measurement of the absolute quantum yields of chemiluminescence and bioluminescence*. *Photochem. Photobiol.* **4**, 1015-1048 (1965).
3. O'Kane, D.J. & Lee, J. Absolute calibration of luminometers with low-level light standards. *Methods Enzymol.* **305**, 87-96 (2000).
4. Cook, P.F. & Cleland, W.W. *Enzyme Kinetics and Mechanism* (Garland Science, 2007).
5. Cleland, W.W. The kinetics of enzyme-catalyzed reactions with two or more substrates or products: III. Prediction of initial velocity and inhibition patterns by inspection. *Biochim. Biophys. Acta(BBA)-Spec. Sect. Enzymol. Subj.* **67**, 188-196 (1963).
6. Guo, Q. et al. Interrogating the activities of conformational deformed enzyme by single-molecule fluorescence-magnetic tweezers microscopy. *Proc. Natl. Acad. Sci.* **112**, 13904-13909 (2015).

REVIEWER COMMENTS

Reviewer #1 (Remarks to the Author):

I am pleased to see that the authors responded to all of the reviewers' inquiries, performed additional experiments, and revised the manuscript. The manuscript is now ready for publication in Nature Communications.

Reviewer #2 (Remarks to the Author):

In this revised version of the manuscript, the authors have clarified diverse points and added additional information to the main text and supplementary materials. The manuscript is thus improved.

I however still struggle with the definition of these measurements as 'single-molecule' experiments:

- The authors measure the emission from ensemble assays (5 nM freely diffusing enzyme in a wide-field microscope with mM concentrations of substrate), at sufficiently high framerate and sensitivity that single photons are detected, similar to doing a per-pixel FCS experiment (of a bulk solution in a cuvette) via single-photon counting.
- This allows them to obtain bulk enzyme properties / kinetics, via fitting to ensemble models. Reaction steps are inferred from these ensemble kinetics. A true single-molecule assay would provide direct stochastic reaction trajectories from individual catalysts.
- The authors demonstrate in Fig. R4 that they are not able to observe reactions from actual single HRP molecules.
- Moreover, they perform experiments on immobilized catalysts in a microscope, so they obtain spatial information (i.e. brightness or # counts), but this is done on catalyst particles, also not single-molecules.

In conclusion, the authors are not analyzing single molecules but single photons emitted stochastically from a large number of molecules.

This should be clarified in the title, the text and the discussion, and the authors should be quite specific what they can and cannot observe. As it is written now, I am convinced that the majority of the readership expect that the paper reports on the observation of luminescence emissions from single HRP molecules over time.

Moreover, I am wondering why the additional experiments (shown in fig. R3 and R4) are not added to the manuscript. They add context and are important controls. For Figure R4: What is the concentration of Amplex red, and is the detection of its fluorescence dependent on the presence of HRP particles?

Reviewer #1 (Remarks to the Author):

I am pleased to see that the authors responded to all of the reviewers' inquiries, performed additional experiments, and revised the manuscript. The manuscript is now ready for publication in Nature Communications.

Response: We thank the reviewer #1 for recommending publication.

Reviewer #2 (Remarks to the Author):

In this revised version of the manuscript, the authors have clarified diverse points and added additional information to the main text and supplementary materials. The manuscript is thus improved. I however still struggle with the definition of these measurements as 'single - molecule' experiments:

- The authors measure the emission from ensemble assays (5 nM freely diffusing enzyme in a wide-field microscope with mM concentrations of substrate), at sufficiently high framerate and sensitivity that single photons are detected, similar to doing a per-pixel FCS experiment (of a bulk solution in a cuvette) via single-photon counting.
- This allows them to obtain bulk enzyme properties / kinetics, via fitting to ensemble models. Reaction steps are inferred from these ensemble kinetics. A true single-molecule assay would provide direct stochastic reaction trajectories from individual catalysts.
- The authors demonstrate in Fig. R4 that they are not able to observe reactions from actual single HRP molecules.
- Moreover, they perform experiments on immobilized catalysts in a microscope, so they obtain spatial information (i.e. brightness or # counts), but this is done on catalyst particles, also not single-molecules.

In conclusion, the authors are not analyzing single molecules but single photons emitted stochastically from a large number of molecules.

This should be clarified in the title, the text and the discussion, and the authors should be quite specific what they can and cannot observe. As it is written now, I am convinced that the majority of the readership expect that the paper reports on the observation of luminescence emissions from single HRP molecules over time.

Moreover, I am wondering why the additional experiments (shown in fig. R3 and R4) are not added to the manuscript. They add context and are important controls. For Figure R4: What is the concentration of Amplex red, and is the detection of its fluorescence dependent on the presence of HRP particles?

Response: We thank the reviewer #2 for acknowledging our efforts in previous round of revision for improving the manuscript. From the comments of this time, reviewer #2 has doubts about the terminology of 'single-molecule' experiments. In this work we develop a single-molecule experiment for detecting single-molecule chemiluminescence (CL) reactions in solution or on catalyst particles which offers spatiotemporal resolution that differs from previous bulk chemiluminescence arrays. To clarify this point raised by the reviewer,

1. First of all, a key point we would like to clarify is that **our single-molecule approach refers to the observation of single-molecule CL reactions, not single HRP molecules.** In order to carry out this, we kept the chemical reaction environment consistent with previous CL studies¹.

² (e.g., free catalyst in solution) and employed spatiotemporal isolation strategy that controls the molecular density (mM concentration of substrate, the same order of magnitude was used in relevant articles about the kinetics of enzyme-catalyzed single-molecule reactions³) as well as the acquisition conditions (high acquisition frequency and high sensitivity detector⁴) for separating individual reactions in both space and time, thereby obtaining a typical ‘on-off’ turnover trajectory, a classic signature of single-molecule signals from stochastic reactions, which is never observed in previous CL reaction studies. In the trajectory, each single photon detected comes from a single CL reaction, and kinetic analysis can be achieved by the counting and distribution of photons, reflecting a statistical result at the single-molecule level. In the ensemble CL experiments, such as the cuvette experiment exemplified by the reviewer #2, researchers can only obtain an average result (e.g., intensity value) of a large number of reactions lacking spatial and temporal resolution information, which is fundamentally different from the statistical and spatiotemporally resolved results in our experiments.

2. **We have a different opinion to the comment that only the detection of single enzyme molecules can be called ‘single-molecule assay’.** Our method has the ability to observe CL reactions at the single-molecule level and this ability is a prerequisite for the future to see CL signals on single enzymes. Direct observation of single enzymes in CL reactions is currently limited by luminescence efficiency of substrate, not limited by the single-molecule sensitivity (single-photon level) in this work (we chose a classical system: HRP and luminol for studying kinetics). To observe the single molecule signals on the immobilized single HRP molecule system, one way is to increase the substrate luminescence efficiency of the chemiluminescent process involving the single enzymes, but this goes out the focus of our current paper.
3. Using our single-molecule measurement, **we have demonstrated the observation of single-molecule CL reactions on single HRP particles (not single HRP molecules)** and single hematin particles. However, reviewer #2 does not recognize the detection of single-molecule CL reactions signals on single particles as single-molecule measurement, which we do not agree. **There are a large number of relevant literatures in the single molecule chemistry/catalysis field demonstrating that single-molecule fluorescence reaction studies on single particles (nanoparticles or microparticles) can be called ‘single-molecule’ studies (and these works do not require the observation of single catalyst molecules), for example,**
 - Xu, W. et al. **Single-molecule** nanocatalysis reveals heterogeneous reaction pathways and catalytic dynamics. *Nat. Mater.* **7**, 992-996 (2008).
 - Andoy, N.M. et al. **Single-molecule** catalysis mapping quantifies site-specific activity and uncovers radial activity gradient on single 2D nanocrystals. *J. Am. Chem. Soc.* **135**, 1845-1852 (2013).
 - Mao, X., Liu, C., Hesari, M. *et al.* Super-resolution imaging of non-fluorescent reactions via competition. *Nat. Chem.* **11**, 687–694 (2019).
 - Mao, X., Chen, P. Inter-facet junction effects on particulate photoelectrodes. *Nat. Mater.* **21**, 331–337 (2022).
 - Chen, T., Zhang, Y. & Xu, W. **Single-molecule** nanocatalysis reveals catalytic activation energy of single nanocatalysts. *J. Am. Chem. Soc.* **138**, 12414-12421 (2016).
 - Xiao, Y. et al. Revealing kinetics of two-electron oxygen reduction reaction at **single-molecule** level. *J. Am. Chem. Soc.* **142**, 13201–13209 (2020).

- Fu, D. et al. Unravelling channel structure–diffusivity relationships in zeolite zsm-5 at the **single-molecule** level. *Angew. Chem., Int. Ed.* **61**, e202114388 (2022).
- Dong, B., Pei, Y., Zhao, F. *et al.* In situ quantitative **single-molecule** study of dynamic catalytic processes in nanoconfinement. *Nat. Catal.* **1**, 135–140 (2018).

It should be noted that our single-molecule CL reaction observation on single catalyst particles is analogous to single-molecule fluorescent reaction observation on single catalyst particles in above large literature (See Fig. R1). As all these analogous experiments can be called single molecule chemistry approaches, we believe the use of 'single-molecule' terminology in our paper is proper and correct.

Fig. R1. Probing single-molecule chemiluminescence reactions on single particles *v.s.* probing single-molecule fluorescence reactions on single particles

We have added the mentioned two experiments shown in our last response letter to Supporting information (Supplementary Fig. 11 and 12). The concentration of amplex red in Fig. R4 is 10 μM . In the presence of HRP, the fluorescence image shows bright spots with flickering in Fig. R4E. In the absence of HRP, the fluorescence has no visible flickering bright spots and ‘on-off’ signals, and only background fluorescence exists, as shown in Supplementary Fig. 12. We thank the reviewer for this suggestion.

In conclusion, we have clarified the use of 'single-molecule' experiment in our work. To avoid any misleading, we have made relevant revisions in the title, text and discussion to clearly demonstrate what we report here.

Revised title in page 1 of manuscript:

‘Direct probing of single-molecule chemiluminescent reaction dynamics’.

Revision in page 8 of manuscript: *‘Using this approach, we are able to observe the single-molecule chemiluminescent reactions in solution with freely diffusing HRP molecules or on single HRP particles (Supplementary Fig. 11). However, we did not manage to observe the single-molecule chemiluminescence reaction emission on single HRP molecules immobilized on a glass slide which is limited by the efficiency of the current chemiluminescence system (see results and discussions in Supplementary Fig. 12).’.*

Revision in page 11 of manuscript: *‘Our approach probes the single-molecule chemiluminescent reaction kinetics in solution with freely diffusing HRP or on single catalyst particles with a single-photon sensitivity down to single-molecule reaction level.’.*

References:

1. Zheng, T. et al. A chemical timer approach to delayed chemiluminescence. *Proc. Natl. Acad. Sci.* **119**, e2207693119 (2022).
2. Whitehead, T.P. et al. Enhanced luminescence procedure for sensitive determination of peroxidase-labelled conjugates in immunoassay. *Nature* **305**, 158-159 (1983).
3. Kou, S.C., Cherayil, B.J., Min, W., English, B.P. & Xie, X.S. Single-Molecule Michaelis–Menten Equations. *J. Phys. Chem. B* **109**, 19068-19081 (2005).
4. Dong, J. et al. Direct imaging of single-molecule electrochemical reactions in solution. *Nature* **596**, 244-249 (2021).

REVIEWERS' COMMENTS

Reviewer #2 (Remarks to the Author):

I thank the authors for the clarifications and the inclusion of the additional data. I have no further concerns and the manuscript is now ready for publication in Nature Communications.